# Magnetic Properties and Microstructure of Ce-Cu-Al Low Melting Alloy Bonding $Sm_2Fe_{17}N_3$ Magnet Fabricated by the Hot-Pressing Method

**Jingwu Zheng** [1,*], **Shitong Yu** [1], **Heng Huang** [1], **Rongyao Li** [1,2], **Wei Cai** [1], **Haibo Chen** [1,2], **Juan Li** [1], **Liang Qiao** [1], **Yao Ying** [1], **Wangchang Li** [1], **Jing Yu** [1] **and Shenglei Che** [1,*]

1   Research Center of Magnetic and Electronic Materials, College of Materials Science and Engineering, Zhejiang University of Technology, Hangzhou 310014, China
2   Hangzhou Chase Technology Co., Ltd., Hangzhou 310000, China
*   Correspondence: zhengjw@zjut.edu.cn (J.Z.); cheshenglei@zjut.edu.cn (S.C.)

**Abstract:** $Sm_2Fe_{17}N_3$ compounds, having excellent intrinsic magnetic properties, are prone to decomposition at high temperatures; thus, a low melting point metal binder is the key to prepare high performance bulk magnets at low temperatures. In this paper, a new low melting point alloy $Ce_{72}Cu_{28-x}Al_x$ was used as the binders, and high-performance Ce-based alloy bonding $Sm_2Fe_{17}N_3$ magnets were realized by the hot-pressing method. The experimental results demonstrated that the content of Al in the Ce-based alloys had an important influence on the performance of the magnets. High performance Sm-Fe-N bonded magnets with remanence of 10.19 KGs and maximum magnetic energy product of 21.06 MGOe were achieved by using 5 wt.% $Ce_{72}Cu_{22}Al_6$ alloy as a binder. At the same time, it was found that the $Ce_{72}Cu_{28-x}Al_x$ alloy has a lower density and better bonding effect than the common Zn binder. Its bonding magnets still have higher performance even with extremely high oxygen content. Therefore, $Ce_{72}Cu_{28-x}Al_x$ alloy with low melting point is a promising new rare earth-based alloy binder. If the oxygen content of the alloy powders could be reduced, higher performance $Sm_2Fe_{17}N_3$ bonded magnets can be prepared.

**Keywords:** $Sm_2Fe_{17}N_3$ bonded magnet; Ce-Cu-Al low melting point alloy; hot-press sintering; rare earth permanent magnet

## 1. Introduction

Permanent magnet materials play vital roles as important functional materials in all aspects of social life. $Nd_2Fe_{14}B$ is widely used as a rare earth magnetic material due to its excellent magnetic properties, however, the low Curie temperature seriously limits its application in high-temperature fields, such as new energy vehicle driving motors [1–4]. To overcome this problem, it is necessary to add less abundant and expensive heavy rare earth elements, such as dysprosium (Dy) and terbium (Tb), which not only further increases the production cost of $Nd_2Fe_{14}B$ magnets but also causes the rare earth reserves of Dy and Tb to be far from meeting their practical application needs [2,5].Therefore, it is very essential to find new rare earth permanent magnet materials that have high Curie temperature with excellent magnetic properties, eliminating the need for heavy rare earth elements.

In recent years, the $Sm_2Fe_{17}N_3$ compound has attracted the attention of researchers due to its excellent intrinsic magnetism, which has comparable saturation magnetization (1.54 T), higher magnetic crystal anisotropy field (14 T), and better corrosion resistance compared with $Nd_2Fe_{14}B$ [6]. More importantly, its Curie temperature is 470 °C, which is much higher than that of the $Nd_2Fe_{14}B$ (312 °C) material [7–10]. Therefore, this material can meet the working temperature requirements of permanent magnet motors in the high-temperature field without adding any heavy rare earth elements. Since its discovery by Coey and Sun et al. [11] in 1990, a large number of scientific research focused on

the development of this new type of permanent magnet material, in order to realize the replacement of $Nd_2Fe_{14}B$ material in the field of high temperature. However, as a metastable compound, $Sm_2Fe_{17}N_3$ will decompose into SmN, $\alpha$-Fe, and $N_2$ at around 600 °C [12]. Moreover, the eutectic temperature (720 °C) of Sm-Fe alloy is higher than the decomposition temperature of $Sm_2Fe_{17}N_3$. Therefore, the Sm-Fe-N magnet can only be fabricated at a low temperature by the solid-phase sintering methods [13], such as the explosive consolidation technique [14–16], compression shearing method [17–20], hot isostatic pressing (HIP) [21,22] and spark plasma sintering method (SPS) [13,23–27].

Since the low melting point grain boundary phase similar to the sintered $Nd_2Fe_{14}B$ cannot be formed during solid-phase sintering, the property of the $Sm_2Fe_{17}N_3$ magnet obtained by the above methods is always unsatisfactory. Therefore, preparing high-density all-metal bulk magnets using low melting point metal or alloy as the binder assisting in low-temperature molding processes can take advantage of the magnetic properties of $Sm_2Fe_{17}N_3$. Among numerous low melting point metals, the addition of Zn can improve the performance of the magnet and its coercivity effectively [28,29]. Otani et al. [30] used 15 vol.% Zn enhancing the coercivity of the Sm-Fe-N magnet from the original 1 kOe to 5.5 kOe. Kuhrt et al. [31] mixed 20 wt.% Zn with magnetic powder and obtained isotropic Zn-bonded magnets with a coercivity of 43.6 kOe and relative density of 80% under the pressing conditions of 425 °C and 270 MPa. Matsuura et al. [32] prepared Zn-bonded Sm-Fe-N magnets with a comprehensive performance through a low-oxygen powder metallurgy process in 2020. However, although the addition of Zn improves the coercivity, it will decrease the saturation magnetization of the $Sm_2Fe_{17}N_3$ magnet [33]. Therefore, in recent years, some researchers have focused on the development of new low melting point alloy binders. In 2018, Otogawa et al. [34] prepared Sm-Fe-Cu-Al quaternary alloy binder with a melting point of 495 °C. Its addition did not affect the saturation magnetization of sintered magnets and showed a better coercivity recovery. Lu et al. [35] from the University of Science and Technology Beijing mixed $Sm_2Fe_{17}N_3$ powder with Sm-Cu nanoflakes, as an oxygen absorber, instead of $Sm_2Fe_{17}N_3$ to provide a low-oxygen sintering environment for the sintering system and the coercivity of the sintered magnet reached 10.3 kOe. These studies demonstrate that Sm-based alloy binders with high activity play a positive role in improving the properties of magnets.

However, although the addition of the Sm-based alloy is of benefit to improve the magnet performance, the melting point of the Sm-based alloy is still relatively high and increases the sintering temperature. Therefore, developing a new rare earth-based alloy binder with similar functions with Sm-based alloys and lower melting points is a promising approach to preparing higher performance $Sm_2Fe_{17}N_3$ bonded magnets. Since the eutectic temperature of the Ce-Cu alloy is much lower than that of the Sm-Cu alloy and both are rare earth-based binary alloys, it may be a good choice to use the Ce-Cu alloy for the development of new low melting point alloy binder. What's more, although the current high-performance $Sm_2Fe_{17}N_3$ magnets are all prepared by the SPS method and realize $Sm_2Fe_{17}N_3$ sintered magnets with excellent performance at the laboratory level, its application in practical production is still difficult. Based on the above analysis, this study uses $Ce_{72}Cu_{28}$ eutectic alloy as the matrix to prepare low melting point $Ce_{72}Cu_{28-x}Al_x$ (x = 0, 3, 6, 9, 12) alloy binders by adding the Al element that is beneficial to the performance improvement of the Nd-Fe-B magnet [36–38]. Meanwhile, high performance Ce-based alloy-bonded $Sm_2Fe_{17}N_3$ magnets were prepared by the hot-press sintering method, which is easier to realize and simple to operate in a practical production than spark plasma sinter. Finally, the magnetic properties and microstructure of the prepared magnets were analyzed.

## 2. Materials and Methods

### 2.1. Material Preparation

The commercial $Sm_2Fe_{17}N_3$ fine powder was produced by a low-oxygen crushing process, supplied by Hangzhou Chease Technology Co., Ltd. (Hangzhou, China). The oxygen content in the equipment was controlled below 10 ppm during crushing to reduce

the degree of oxidation of the powder. $Ce_{72}Cu_{28-x}Al_x$ (x = 0, 3, 6, 9, 12) alloy ingots were obtained by arc melting of high-purity Ce ingots (99.5%), Cu grains (99.9%) and Al grains (99.9%) under Ar atmosphere. The alloy ingots were rapidly quenched by the melt-spinning method at a rotational speed of 55 m/s to obtain the Ce-based alloy strip casting. In the rapid quenching process, to reduce the degree of oxidation of the thin strip, the cavity of equipment was quenched with inert gas several times through ventilation to reduce the oxygen content. After preliminarily crushing the casting strip in the glove box, it was further crushed by a planetary ball miller for 8 h and passed through a 500-mesh sieve to obtain $Ce_{72}Cu_{28-x}Al_x$ fine alloy powder. During ball milling, the n-hexane was used as the solvent for wet milling, the ratio of ball to the material was 20:1, and the rotational speed was 200 rpm.

The $Sm_2Fe_{17}N_3$ fine powder was mixed with 5 wt.% $Ce_{72}Cu_{28-x}Al_x$ (x = 0, 3, 6, 9, 12) alloy powder in a high vacuum glove box with the oxygen content lower than 1 ppm to obtain 5 wt.% $Ce_{72}Cu_{28-x}Al_x/Sm_2Fe_{17}N_3$ mixed powder. Then, 5 wt.% $Ce_{72}Cu_{28-x}Al_x/Sm_2Fe_{17}N_3$ green compacts were prepared by orientation pressing under a 2T magnetic field and cold isostatic pressing under 500 MPa pressure. After that, the green compacts were placed in a mold with an inner diameter of 10 mm, preheated in heating device for 1 min and kept under the pressure of 2 GPa for 1.5 min. Finally, the all-metal $Sm_2Fe_{17}N_3$ magnets bonded by Ce-Cu-Al low melting alloys were obtained after demolding. Due to certain temperature deviation and heat losses, the hot-press sintering temperature selected in this experiment was 495 °C.

### 2.2. Performance Characterization

The melting point of the $Ce_{72}Cu_{28-x}Al_x$ alloy was determined by a differential scanning calorimeter (DSC: TA Q2000) at a heating rate of 10 °C/min using $N_2$ as the protective gas. The density and magnetic performance of the alloys were measured by the Archimedes method and vibrating sample magnetometer (VSM: LakeShore 7404), respectively. The oxygen content of the $Sm_2Fe_{17}N_3$ magnetic powder and $Ce_{72}Cu_{28-x}Al$ alloy powder was determined by oxygen, nitrogen, and hydrogen elemental analyzers (elemental analyzers: Elementrac ONH-p). To prevent the powder from oxidation, the sample preparation was carried out in the glove box. The magnetic properties test of the hot-pressed magnets was done at the ATM-4 magnetic measuring instrument, and the density was calculated by the volume measurement method. The microstructural features of powder, magnet cross-sections and elemental distribution of polished surfaces were observed by scanning electron microscope with an energy spectrum. (Desktop SEM: Phenom ProX, Pro, and Pure G6).

### 3. Results and Discussion

The particle size distribution and micro-morphology of $Sm_2Fe_{17}N_3$ fine powder obtained by the low-oxygen crushing of commercial coarse powder are shown in Figure 1a,b, respectively. The average particle size of the powder is around 4.42 μm with an oxygen content of 4781.54 ppm. Figure 2a is the outline drawing of the obtained ingot after arc melting $Ce_{72}Cu_{28-x}Al_x$ (x = 0, 3, 6, 9, 12) alloy. Since the oxygen content inside the furnace is extremely low and protected by argon gas, the obtained ingot shows a small degree of oxidation and exhibits silvery-white metallic luster. Figure 2b shows the alloy strips obtained after the rapid quenching of ingots by the melt-spinning method at a rotational speed of 55 m/s. Different from other metals, the Ce-Cu-Al alloy rapid quenching strips to demonstrate well toughness, and strips more than 10 cm often appear after stripping. In addition, the rapid quenching strips are easily oxidized in the air due to the high content of rare earth metal in the alloy; therefore, it is stored in hexane to reduce the degree of oxidation. Figure 2c shows the powder morphology of the strip casting after ball milling and sieving. The changes in oxygen content of the alloy thin strips and powders after ball milling are shown in Table 1. By strictly controlling the oxygen content, the oxygen content of the alloy thin strips is kept below 1500 ppm. However, since the rare earth content in the

$Ce_{72}Cu_{28-x}Al_x$ alloy is above 85 wt.%, even if the organic solvent is used during ball milling and the subsequent operations are carried out in high vacuum glove box, the powders after ball milling still undergo violent oxidation, resulting in higher oxygen content of the powders (70,000 ppm). Fortunately, even with such a high oxygen content, the alloy powders still show very high reactivity, therefore, it will spontaneously ignite due to intense air oxidation.

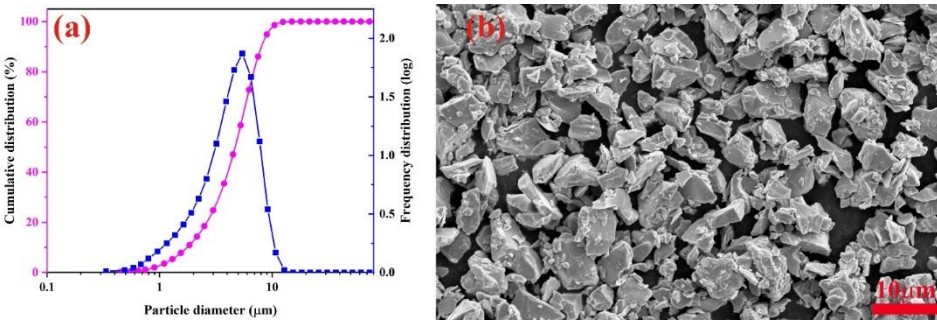

**Figure 1.** (**a**) Particle size distribution and (**b**) microscopic morphology of $Sm_2Fe_{17}N_3$ fine powder.

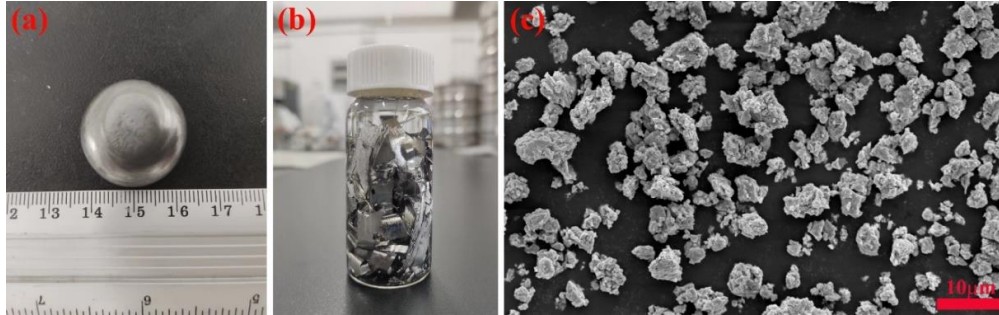

**Figure 2.** (**a**) Appearance of $Ce_{72}Cu_{28-x}Al_x$ alloy ingot, (**b**) rapid quenching strip, and (**c**) micro structure morphology of alloy powder.

**Table 1.** Oxygen content (ppm) summary of $Ce_{72}Cu_{28-x}Al_x$ alloy strip and powders after ball milling.

| $Ce_{72}Cu_{28-x}Al_x$ | | x = 0 | x = 3 | x = 6 | x = 9 | x = 12 |
|---|---|---|---|---|---|---|
| Oxygen content/ppm | quenching strips | 1319.79 | 1225.04 | 1466.27 | 1207.89 | 1194.59 |
| | powders | 85,399.27 | 97,532.32 | 90,098.35 | 82,642.71 | 73,331.51 |

The melting point is an important index to evaluate the adhesive properties of the alloys and its level can determine the optimal sintering temperature of the magnet; therefore, the thermodynamic behavior of the alloy is an extremely important factor. The DSC curve of $Ce_{72}Cu_{28-x}Al_x$ alloys with the changing Al content is shown in Figure 3. Upwards peaks indicate exothermic and downwards peaks reveal endothermic reactions. When x = 0, the melting point of $Ce_{72}Cu_{28}$ binary alloy is 426.34 °C. After adding Al, the melting point of $Ce_{72}Cu_{28-x}Al_x$ (x = 3, 6, 9, 12) ternary alloy decreases and is lower than Zn (419.53 °C) [27,28], and the lowest temperature reaches up to 389.40 °C for $Ce_{72}Cu_{22}Al_6$ alloy. However, with the increase in Al content, the melting point does not change much and is maintained at around 390 °C. In addition, it can be found from the curve that the $Ce_{72}Cu_{25}Al_3$ alloy has two endothermic peaks at 387.50 °C and 413.43 °C, which may be attributed to the addition of Al resulting in a deviation from the eutectic composition [39]. Compared with the Sm-Cu binary eutectic alloy (melting point = 555 °C) and Sm-Fe-Cu-Al alloy binder (melting point = 495 °C), $Ce_{72}Cu_{28-x}Al_x$ alloys have a lower melting point. Under the same preparation process, its addition can greatly reduce the forming temperature of the $Sm_2Fe_{17}N_3$ magnet and achieve the fabrication of the Sm-Fe-N bonded magnet at a lower temperature.

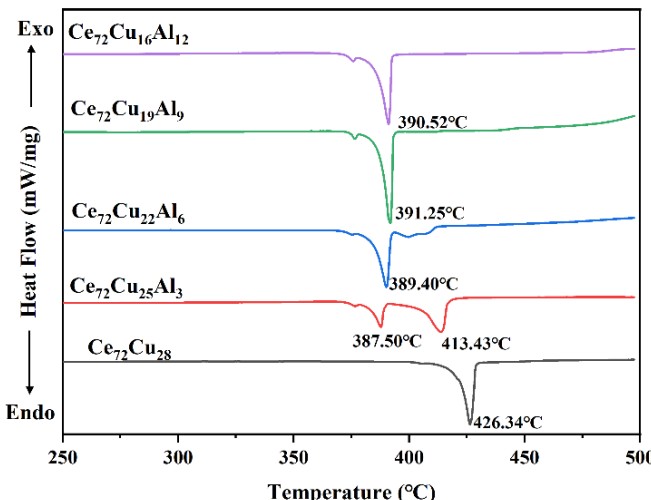

**Figure 3.** DSC curves of $Ce_{72}Cu_{28-x}Al_x$ alloys with different Al contents.

The alloy binder is mostly located at the grain boundaries of the magnet and the magnetic performances of the grain boundary phase influence the final properties of the magnet. Thus, the magnetic performances of $Ce_{72}Cu_{28-x}Al_x$ alloy powders were also tested by vibrating the sample magnetometer, and its VSM and local amplification diagrams are shown in Figure 4. Both Ce-Cu binary alloy and Ce-Cu-Al ternary alloys demonstrate weak magnetic characteristics and the magnetic performances of the alloys change with the increase in Al content. When the atomic percentage of Al increases from x = 0 at.% to x = 9 at.%, the remanence of the alloys gradually increases from 0.53 emu/g to 3.01 emu/g. Then, when it increases to 12 at.%, the remanence of the $Ce_{72}Cu_{16}Al_{12}$ alloy decreases sharply, even lower than that of $Ce_{72}Cu_{28}$ alloy. Coercivity also shows a similar variation trend. After adding Al, the coercivity of the powders rapidly increases from 1304.57Oe at x = 0 at.% to more than 6000 Oe and reaches 7059.16 Oe at x= 3 at.%. However, when Al content increases to 12 at.%, the coercivity of the $Ce_{72}Cu_{16}Al_{12}$ alloy decreases rapidly to several hundred Oe. These results suggest that the Ce-Cu binary alloy exhibits weak magnetic properties, and the addition of Al demonstrates a positive effect on its magnetic performance; therefore, the coercivity of Ce-Cu-Al alloy powders greatly increases while the remanence is also improved. However, when the Al content is too high, the remanence and coercivity will drop sharply.

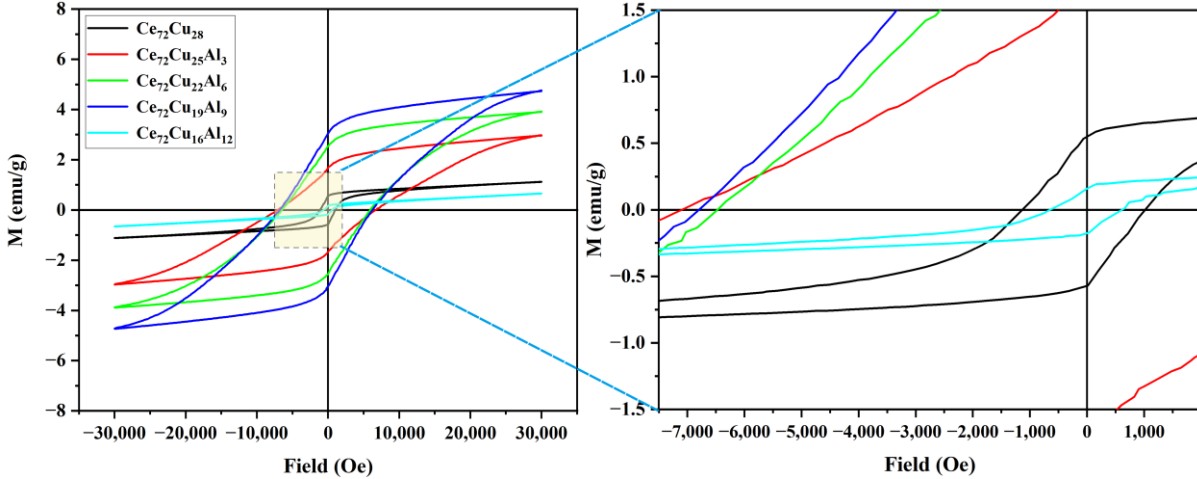

**Figure 4.** VSM diagram of $Ce_{72}Cu_{28-x}Al_x$ alloys and their partial magnification.

To determine the effect of Ce-based alloy binders with low melting points and weak magnetic characteristics on the magnetic properties of the Sm-Fe-N magnet, the

$Ce_{72}Cu_{28-x}Al_x/Sm_2Fe_{17}N$ green compacts with 5 wt.% alloy addition were consolidated by the hot-press sintering method. The appearance, density and magnetic properties of the bonded magnets are shown in Figure 5. With the increase in Al content, the density of the magnets decreases gradually, from 6.66 $g/cm^3$ at x = 0 at.% to 6.28 $g/cm^3$ at x = 12 at.%. In contrast, the remanence, coercivity, and maximum magnetic energy product increase first and then decrease, reaching the maximum values of 10.12 kGs, 5.63 kOe, and 21.06 MGOe when the atomic percentage of Al is 6 at.%.

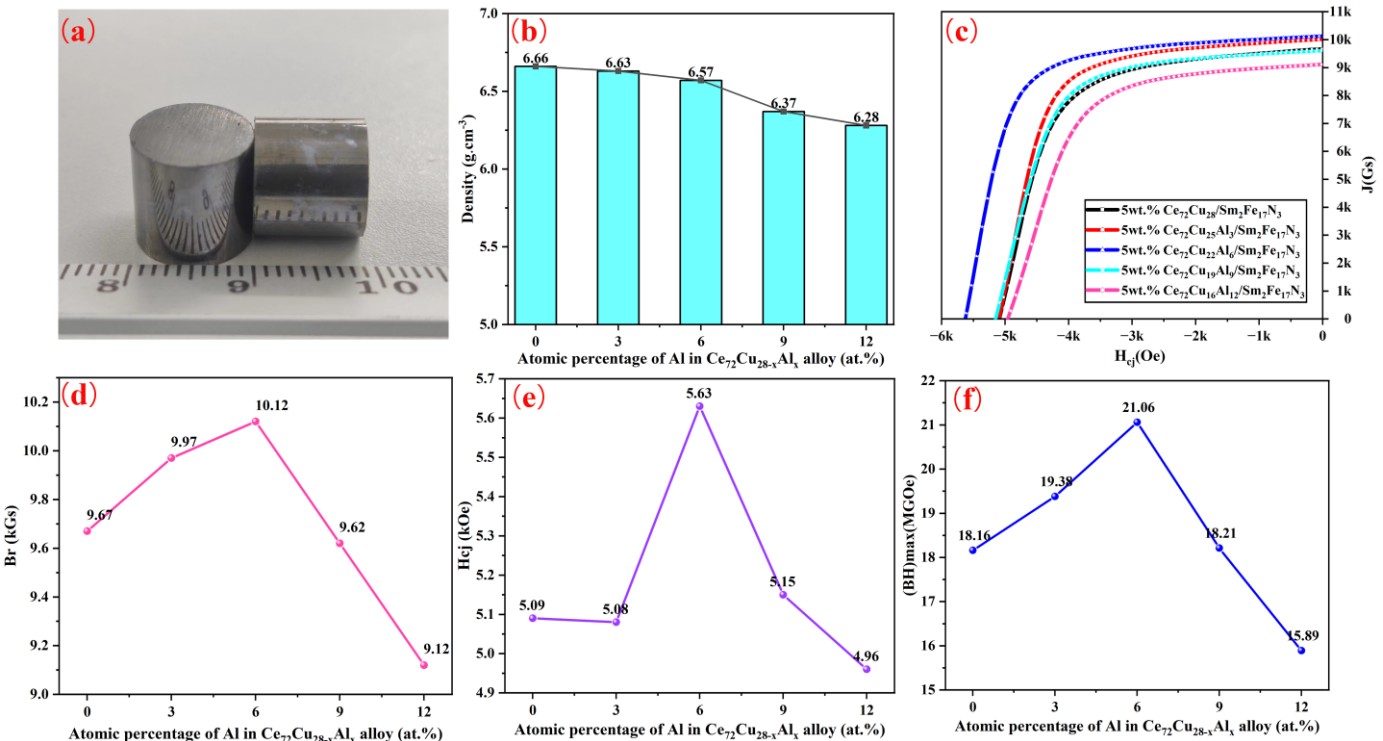

**Figure 5.** Change in properties of Sm-Fe-N hot-pressing magnets by adding 5 wt.% $Ce_{72}Cu_{28-x}Al_x$ with different Al contents and (**a**) appearance (**b**) density, (**c**) demagnetization curves, (**d**) remanence, (**e**) intrinsic coercivity, and (**f**) maximum energy product of magnets.

The decrease in magnet density is related to the change in density of $Ce_{72}Cu_{28-x}Al_x$ (x = 0, 3, 6, 9, 12) alloys. Figure 6 shows the variation in density of $Ce_{72}Cu_{28-x}Al_x$ (x = 0, 3, 6, 9, 12) alloys with the increase in Al content. The ordinate value corresponding to the red dashed line is 7.14 $g/cm^3$, equivalent to the density of Zn [2]. The density of $Ce_{72}Cu_{28-x}Al_x$ alloys is lower than that of then Zn metal [28]. When the atomic percentage of Al is 0 at.%, the density of $Ce_{72}Cu_{28}$ binary alloy is 7.01 $g/cm^3$. With the increase in Al content, the alloys' density decreases gradually, reaching up to 6.71 $g/cm^3$ at x = 12 at.%. Therefore, the density of the $Sm_2Fe_{17}N_3$ bonded magnets decrease gradually with the addition of Ce-based alloy binders under the same experimental conditions. At the same time, comparing the density of the Ce-based alloy and Zn, it is also demonstrated that under the same mass fraction, Ce-based alloy powders have a larger volume fraction, which is more conducive to the densification of the magnet after mixing them with magnetic powder [29,32].

According to the magnetism theory, the remanence is positively related to the density of the magnet. Therefore, under the same conditions, the remanence should show the same variation trend to density. However, in this study, the variation in the remanence of the Ce-based alloy bonded magnet is different from that of the magnets' density, which does not demonstrate a gradually decreasing trend, reaching the maximum value at x = 6 at.%. The possible reason for this phenomenon is that among $Ce_{72}Cu_{28-x}Al_x$ (x = 0, 3, 6, 9, 12) alloys, $Ce_{72}Cu_{22}Al_6$ alloy has the most suitable bonding effect, resulting in better bonding between magnetic powders. Therefore, even though the density of its bonding magnet

is lower than that of the $Ce_{72}Cu_{28}$ and $Ce_{72}Cu_{25}Al_3$ bonding magnets, the remanence is the highest.

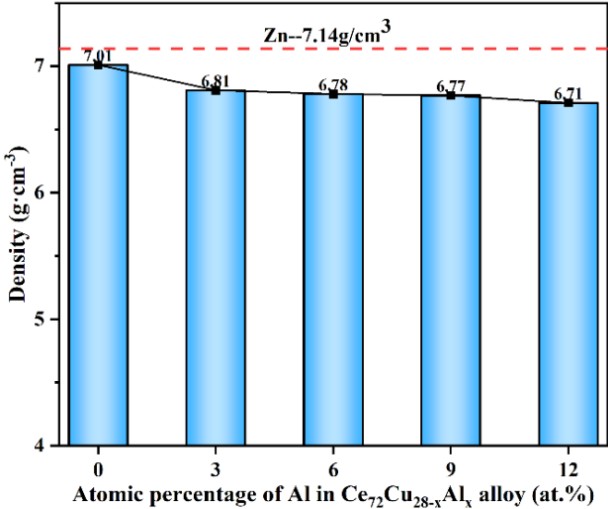

**Figure 6.** Density variation of $Ce_{72}Cu_{28-x}Al_x$ alloys with the change in Al contents.

In addition, it can be found from Figure 5e that the coercivity of the $Ce_{72}Cu_{22}Al_6$ alloy bonding magnet is higher than that of the other samples. Regarding the reason that only $Ce_{72}Cu_{22}Al_6$ alloy bonding magnets have higher coercivity, on the one hand, it may be because the $Ce_{72}Cu_{22}Al_6$ alloy has the lowest melting point among all alloys, which is beneficial to the bonding between magnetic particles during the magnet preparation process. On the other hand, by comparing the remanence and density variation of the magnets, it can be observed that even though the density of the $Ce_{72}Cu_{22}Al_6$ bonding magnets did not reach the highest, its remanence is indeed the highest among all magnets, indicating that $Ce_{72}Cu_{22}Al_6$ has the best wetting ability on magnetic powders among all alloys and is more helpful to prevent the magnetic exchange coupling effect between magnetic particles. Therefore, the above two points are perhaps the main reasons why the coercivity of $Ce_{72}Cu_{22}Al_6$ alloy bonding magnets is higher than other samples.

However, although the coercivity of the $Ce_{72}Cu_{22}Al_6$ alloy bonding magnet reaches its highest value of 5.63 kOe, the coercivity is still at a low level. The reason for the low coercivity is attributed to the oxygen content of the Ce-based alloy powder. Since the oxygen content of the raw powders have a great influence on the coercivity of the final magnet, controlling it well is an effective method to improving the coercivity, so current generation in high coercivity Sm-Fe-N magnet strongly relies on the low-oxygen powder metallurgy process [32,40,41]. Although the oxygen content of Sm-Fe-N magnetic powder in this study is low, the oxygen content of the Ce-based alloy binders rises rapidly after ball milling, thus, sharply increasing the oxygen content of the sintered magnet and eventually leading to a decrease in the coercivity of the magnet. Therefore, reducing the oxygen content of the alloy powders will help to improve its wetting effect on the magnetic powders increasing the magnet density and further prevent the magnetic exchange coupling effect between magnetic powders improving the magnet coercivity.

Since the current high-performance $Sm_2Fe_{17}N_3$ bonded magnets mainly use Zn as the binder, and the $Ce_{72}Cu_{28-x}Al_x$ alloy-bonded magnets show the highest performance at $x = 6$ at.%, it is necessary to compare differences in magnetic properties and microstructure. Table 2 lists the performance comparison between the Zn and $Ce_{72}Cu_{22}Al_6$ alloy-bonded magnets under the same experimental conditions. Since Zn cannot reach the same oxygen content as Ce-based alloy powder, low oxygen and high oxygen content Zn powders with 6436.33 ppm and 13,109.93 ppm, respectively, are selected for comparison [27]. The magnet using low oxygen content Zn powder as the binder shows the highest performance. However, when the oxygen content of Zn powder increases from 6346.33 ppm to

13,109.93 ppm, the coercivity of the bonded magnet decreases rapidly from 6.16 KOe to 5.17 KOe, and the maximum magnetic energy product also decreases from 22.77MGOe to 20.14 MGOe. In contrast, although the oxygen content of $Ce_{72}Cu_{22}Al_6$ alloy powder is seven times higher than that of high-oxygen Zn powder, the coercivity and maximum magnetic energy product of the alloy bonding magnet is higher than that of high oxygen Zn powder bonding magnet, and slightly lower than that of the hot-pressing magnet bonded by low oxygen Zn powder. In addition, Since the $H_k/H_{cj}$ value of the Ce-based alloy bonding magnet is higher than those of Zn bonding magnets when $Ce_{72}Cu_{22}Al_6$ alloy is used as binder, squareness of the demagnetization curve is higher than that of the Zn bonding magnet, indicating that the Ce-Cu-Al alloy bonding magnet has a stronger ability to resist the interference of external magnetic field and environmental factors under dynamic working conditions and has a better stability. Therefore, compared with Zn, $Ce_{72}Cu_{22}Al_6$ alloy with high activity and weak magnetic characteristics has obvious advantages as a binder in the Sm-Fe-N magnet. If the oxygen content of alloy powders can be reduced, the performance of the $Sm_2Fe_{17}N_3$ bonded magnet may higher than that of the Zn bonding magnet under the same conditions. The comparison of the cross-sectional morphology of bonded magnets with the same amount of Zn and $Ce_{72}Cu_{22}Al_6$ alloy is shown in Figure 7. When Zn is used as the binder, although it shows a certain bonding effect, there are still a large number of holes in the cross-section. Moreover, due to the uneven distribution of Zn powder, some magnetic powder particles are only squeezed together under high pressure, without any mutual bonding effect. In contrast, when $Ce_{72}Cu_{22}Al_6$ alloy powder with a lower melting point and density is used as binder, only fewer holes in the cross-section of the magnet are observed and the bonding between magnetic particles is more closely than that of Zn. In addition, a piece of the area where the alloy is cooled and solidified after melting is observed in the cross-section of the magnet, and the magnetic particles in this region are surrounded by the alloy, further indicating better fluidity and better bonding effect of Ce-based alloy than Zn metal. Therefore, in terms of magnet performance, even though the Ce-based alloy binders have a very high oxygen content, the remanence of the bonded magnets can still reach a level comparable to that of the Zn bonded magnets.

**Table 2.** Performance comparison of 5 wt.% $Ce_{72}Cu_{22}Al_6$ alloy and Zn with different oxygen content bonded magnets.

|  | Oxygen Content of Binder (ppm) | $B_r$ (kGs) | $H_{cj}$ (kOe) | $(BH)_{max}$ (MGOe) | $H_k/H_{cj}$ (%) |
|---|---|---|---|---|---|
| Zn | 6436.33 | 10.37 | 6.16 | 22.77 | 72.60 |
| Zn | 13,109.93 | 10.24 | 5.17 | 20.14 | 67.90 |
| $Ce_{72}Cu_{22}Al_6$ | 90,098.35 | 10.12 | 5.63 | 21.06 | 74.10 |

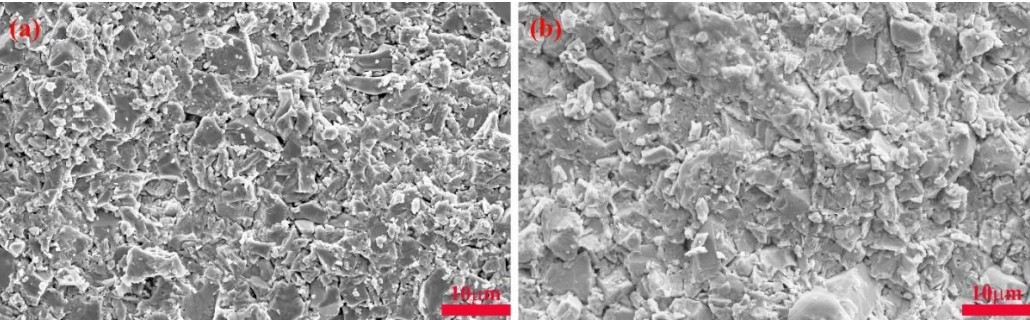

**Figure 7.** The cross-sectional morphology comparison of Zn and $Ce_{72}Cu_{22}Al_6$ alloy bonded Sm-Fe-N magnets with the addition of 5 wt.% of Al. Cross-sectional morphology of (**a**) Zn bonded magnet and (**b**) $Ce_{72}Cu_{22}Al_6$ bonded magnet.

The backscattering diagram and EDS energy spectrum of the polished section of the bonded $Ce_{72}Cu_{22}Al_6$ alloy magnet are shown in Figure 8. Region 1 is the main phase of $Sm_2Fe_{17}N_3$, region 2 is the $Ce_{72}Cu_{22}Al_6$ alloy adhesive (the white color is due to its relatively large molecular weight), and region 3 is a small amount of Sm-rich phase. Since there are holes in the magnet and the sample is prone to oxidation, black spots are observed in the backscatter pattern. Different from the Ce-based alloy powder with a smaller particle size after ball milling, the Ce-based alloy has a larger deformation and shows sign of flow between the magnetic powders after hot-pressure sintering achieving the well mutual bonding between magnetic powder particles. What is more, Ce-based alloy can penetrate better into the magnetic powders, preventing the magnetic exchange coupling effect between the powders that is beneficial for the improvement of the coercivity. Therefore, the coercivity of the $Ce_{72}Cu_{22}Al_6$ alloy bonding magnets is higher than that of the high oxygen content Zn-bonded magnets. If the oxygen content of the Ce-based alloy powders could be better controlled, it will be very helpful for improvement the coercivity of the $Sm_2Fe_{17}N_3$-bonded magnet.

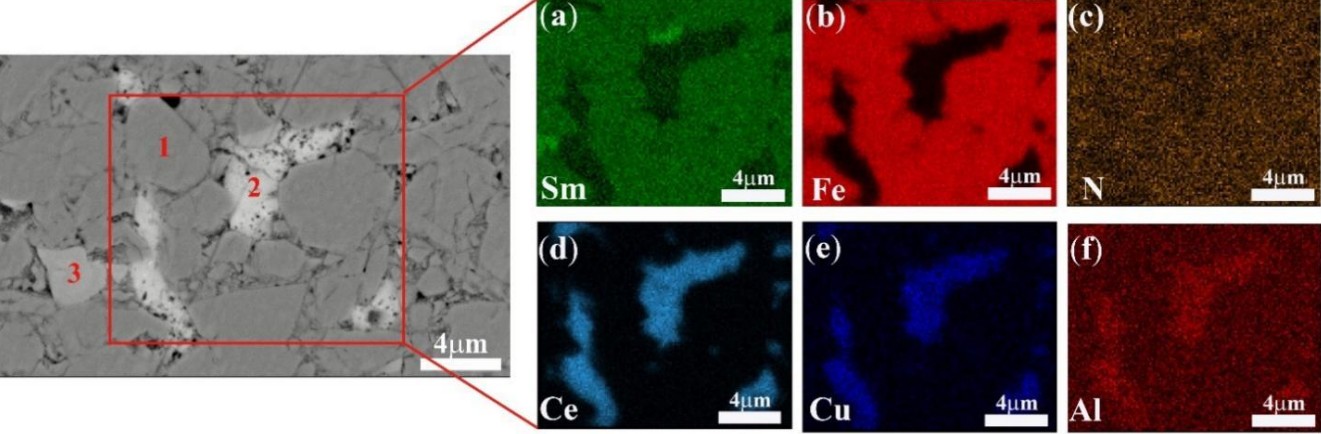

**Figure 8.** EDS spectra of a polished cross-section from Sm-Fe-N bonded magnet using $Ce_{72}Cu_{22}Al_6$ powder binder and elemental distribution maps of (**a**) Sm, (**b**) Fe, (**c**) N, (**d**) Ce, (**e**) Cu, and (**f**) Al.

## 4. Conclusions

In this study, the $Ce_{72}Cu_{28-x}Al_x$ (x = 0, 3, 6, 9, 12) alloys with a lower melting point, higher activity, and better bonding effect were prepared and high-performance $Sm_2Fe_{17}N_3$-bonded magnets were fabricated by the hot-press sintering method. The experimental results demonstrated that the $Ce_{72}Cu_{28-x}Al_x$ alloys with a lower density and weak magnetic properties were more conducive to the fabrication of $Sm_2Fe_{17}N_3$ bonded magnets under high pressure and the improvement of magnet performance. What is more, despite the alloy powder having a very high oxygen content, the performance of its bonding magnets was better than that of the high oxygen content Zn powder bonding magnet. Therefore, it is more advantageous to use high-activity Ce-based alloy powder as a binder than Zn powder. If the oxygen content of alloy powder could be reduced, the $Sm_2Fe_{17}N_3$ bonded magnet with higher performance can be realized. Finally, when 5 wt.% $Ce_{72}Cu_{22}Al_6$ alloy powder was used as binder, the high-performance $Sm_2Fe_{17}N_3$-bonded magnet with remanence, coercivity, and the maximum magnetic energy product of 10.12 KGs, 5.63 KOe, and 21.06 MGOe was obtained, respectively.

**Author Contributions:** Conceptualization, J.Z. and S.C.; methodology, S.Y. and J.Z.; investigation, S.Y., H.H. and R.L.; writing—original draft preparation, S.Y.; writing—review and editing, S.Y. and J.Z.; supervision, J.L., L.Q., Y.Y., W.L. and J.Y.; funding acquisition, J.Z. and W.C. and H.C. and S.C. All authors have read and agreed to the published version of the manuscript.

**Funding:** This work was financially sponsored by Basic public welfare research program of Zhejiang Province (funder: J.Z.; funding number: LGG22E010010) and National Natural Science Foundation of China (funder: H.C.; funding number: 52101235) and Key R&D Project of Zhejiang Provincial Department Science and Technology (funder: S.C.; funding number: 2021C01172) and Scientific Research Fund of Zhejiang Provincial Education Department (funder: W.C.; funding number: Y202043346).

**Institutional Review Board Statement:** Not applicable.

**Informed Consent Statement:** Not applicable.

**Data Availability Statement:** Not applicable.

**Conflicts of Interest:** The authors declare no conflict of interest.

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
