# Peer review of "Magnetic Properties and Microstructure of Ce-Cu-Al Low Melting Alloy Bonding Sm2Fe17N3 Magnet Fabricated by the Hot-Pressing Method"

_magnetochemistry, doi:10.3390/magnetochemistry8110149_

Round 1

Reviewer 1 Report

The Sm2Fe17N3 compound has become a high temperature permanent magnet promising material. The result of nitrogenation gives a substantially higher Curie temperature, higher saturation magnetization, remanence and coercivity. It is therefore a candidate material for the development of new high performance permanent magnets for high temperature applications. The authors uses low melting point Ce72Cu28-xAlx (x=0, 3, 6, 9, 12) alloy binders in order to improve to performance improvement of the Nd-Fe-B magnet. High performance Ce-based alloy bonded Sm2Fe17N3 magnets were prepared by hot-press sintering method. Finally, the magnetic properties and microstructure of the prepared magnets were analyzed. The work seems relevant in the idea of proposing alternatives in order to improve the performance of permanent magnets based on rare earths, even knowing that there are several researches that aim to seek alternatives to rare earths.

comments and questions below:

1. The abstract of the work can be rewritten. Zn based magnets as a low melting point metal is not a result of the work, it is only used to compare with Ce-based alloy bonded Sm2Fe17N3 magnets obtained in the work.

2. The authors present the results and analyzes based on the comparison with magnets based on Zn, however it is necessary to place references on these results, for example in lines 173, 222, 223, 228, 259, 261, 286, etc. . In addition to the values in table 2.

3. In addition to the composition of permanent magnets, the magnetic properties also depend on the microstructure and domain structure. Therefore, in the manufacture of permanent magnets, it is a combination of the chemical elements present together with the microstructure produced by thermal and mechanical processing that ultimately determines the magnetic properties. Could the authors better explain the relationship between magnetic properties and microstructure in their systems? There is some explanation in the text, but it is focused on comparing with Zn-based systems.

4. Ce72Cu28-xAlx alloy magnets have the highest performance at x=6 at. %, could you explain how oxygen content affects magnetic properties and performance?

In my opinion, the manuscript can be considered for publication after minor revision.

Reviewer 2 Report

The authors of the manuscript titled “Magnetic properties and microstructure of Ce-Cu-Al low melting alloy bonding Sm2Fe17N3 magnet fabricated by the hot-pressing method” have presented the objectives along with the results and discussions very clearly. There are a very few minor clarifications in the manuscript.

1.       In line 268 what do the authors mean by square degree of the magnet?

2.       The authors should include a picture of the fabricated magnet.

3.       In fig.5, why does the coercivity of the just one sample higher compared to others? Is there an explanation for such behavior?

4.       What is the error range in the Br and Hc,i values derived from hysteresis graphs in figure 5?

5.       How were the samples prepared for the SEM analysis? The question arises as there may be oxidation during sample preparation procedure
